# Cushing’s Disease in the Animal Kingdom: Translational Insights for Human Medicine

**DOI:** 10.3390/ijms26178626

**Published:** 2025-09-04

**Authors:** Elena Massardi, Germano Gaudenzi, Silvia Carra, Monica Oldani, Ilona Rybinska, Luca Persani, Giovanni Vitale

**Affiliations:** 1Department of Medical Biotechnology and Translational Medicine, University of Milan, 20129 Milan, Italy; elena.massardi96@gmail.com (E.M.); ilona.rybinska@unimi.it (I.R.); luca.persani@unimi.it (L.P.); 2Laboratory of Geriatric and Oncologic Neuroendocrinology Research, IRCCS, Istituto Auxologico Italiano, Cusano Milanino, 20095 Milan, Italy; g.gaudenzi@auxologico.it (G.G.); monicaoldani12@gmail.com (M.O.); 3Laboratory of Endocrine and Metabolic Research, IRCCS, Istituto Auxologico Italiano, Cusano Milanino, 20095 Milan, Italy; s.carra@auxologico.it

**Keywords:** Cushing’s disease, hypothalamic–pituitary–adrenal axis, dog, horse, cat, mouse, zebrafish

## Abstract

Cushing’s disease (CD) is a rare neuroendocrine disorder caused by ACTH-secreting pituitary adenomas, presenting significant diagnostic and therapeutic challenges. Given the evolutionary conservation of the hypothalamic–pituitary–adrenal axis, this review explores the translational value of spontaneous CD forms in dogs, horses, cats, small mammals, and rats, as well as of experimental models in mice, rats, and zebrafish. Dogs are the most studied, showing strong molecular and clinical similarities with human CD, making them valuable for preclinical drug and diagnostic research. While equine and feline CD are less characterized, they may provide insights into dopaminergic therapies and glucocorticoid resistance. Nevertheless, practical and ethical challenges limit the experimental use of companion animals. In preclinical research, mouse models are widely used to study hypercortisolism and test therapeutic agents via transgenic and xenograft strategies. Conversely, few studies are available on a zebrafish transgenic model for CD, displaying pituitary corticotroph expansion and partial resistance to glucocorticoid-negative feedback at the larval stage, while adults exhibit hypercortisolism resembling the human phenotype. Future transplantable systems in zebrafish may overcome several limitations observed in mice, supporting CD research. Collectively, these animal models, each offering unique advantages and limitations, provide a diverse toolkit for advancing CD research and improving human clinical outcomes.

## 1. Introduction

Cushing’s disease (CD) was named after Harvey Cushing, the American neurosurgeon who first described this human neuroendocrine pathological condition in 1932. CD arises from a tumor that autonomously and continuously secretes the adrenocorticotropic hormone (ACTH), thereby impairing the hypothalamic–pituitary–adrenal (HPA) axis. The resulting loss of the circadian cortisol production and the persistently elevated cortisol level lead to the peculiar CD clinical signs [1]. The possibility to perform preclinical studies on animal models that recapitulate CD clinical and molecular features provides a unique opportunity to gain insights into its pathogenic mechanisms, investigate the systemic effects of ACTH and glucocorticoids (GCs) deregulation, and develop new therapeutic strategies, especially given the lack of human ACTH-producing cell lines for in vitro research [2]. Due to the high evolutionary conservation of the HPA axis in vertebrates [3], spontaneous pathological conditions resembling CD have been documented in various animal species, including dogs, horses, cats, small mammals, and rats. In this frame, several studies aimed to determine whether the characterization of these CD forms might have translational implications for human patients or hold solely veterinary significance. In this review, we summarize these studies alongside those conducted in classical preclinical research species, such as mice, rats, and zebrafish, in which stable and transient CD models have been developed over the years.

A comprehensive literature search was conducted using PubMed as the primary online database. The search was performed without temporal restrictions and employed a combination of keywords, including “Cushing’s disease”, “hyperadrenocorticism”, “pituitary adenoma”, “dog”, “horse”, “cat”, “small mammals”, “rat”, “mouse”, and “zebrafish.” Articles were included if they were written in English and published in peer-reviewed journals. Studies were excluded if they were irrelevant to the topic, duplicated, or published in languages other than English.

## 2. CD in Humans

Human CD, which accounts for approximately 70% of Cushing’s syndrome cases, is considered a rare endocrine disorder [4], with an estimated incidence of 0.24 per 100,000 cases with a 95% confidence interval, as reported in a recent systematic review. The work included thirteen studies, demonstrating CD epidemiology to be similarly reported across different areas worldwide, with some exceptions related to regional differences or observation period intervals [5].

CD is caused by ACTH-secreting tumors that exhibit partial resistance to GCs. These tumors originate primarily from the anterior lobe of the pituitary gland, with only 5–10% of cases arising from extra-pituitary tissues [6,7]. Only isolated case reports have described CD arising from tumors secreting corticotropin-releasing hormone (CRH), which is normally produced by the hypothalamus to control the expression of ACTH in the pituitary gland, or co-secreting ACTH and CRH [8,9,10,11,12]. The peculiar chronic hypercortisolism state of CD patients imposes a significant clinical burden, leading to an increased prevalence of comorbidities, diminished health-related quality of life, and an increased risk of mortality. Patients with CD frequently exhibit characteristic physical features and complications, summarized in Table 1. Given the broad spectrum of clinical manifestations, timely recognition and management of CD are crucial to reducing morbidity and improving patient outcomes [1].

Surgical removal of ACTH-secreting tumors remains the preferred treatment for CD. However, the procedure carries a high risk of recurrence, and the long-term physical and psychosocial consequences of chronic hypercortisolism may persist even in patients considered cured. For cases of persistent or recurrent disease, current therapeutic approaches include repeat surgery, radiotherapy, or bilateral adrenalectomy. Currently available pharmacological options for managing these patients encompass different strategies, including the decrease in ACTH secretion at the hypothalamic–pituitary level (neuromodulatory agents such as somatostatin analogs and dopamine agonists), the inhibition of cortisol synthesis at the adrenal level (steroidogenesis inhibitors), and the blockade of cortisol action via glucocorticoid receptor antagonists (i.e., mifepristone) [1,13,14,15,16,17].

## 3. Spontaneous Models of CD in Animals

In veterinary medicine, studies on spontaneous forms of CD often refer to this condition using the broader term hyperadrenocorticism [18]. This terminology encompasses the overproduction of any adrenocortical hormone, as seen in Cushing’s syndrome, without necessarily distinguishing between ACTH-dependent and -independent forms. To address this limitation, many reports adopt specific terminology to refer to forms that more closely align with the human definition of CD. Spontaneous CD models are summarized in Table 2.

### 3.1. Dog (Canis lupus familiaris)

Canine CD, often referred to as pituitary-dependent hyperadrenocorticism (PDH), is one of the most common endocrinological disorders in this species (1 to 2 cases/1000 dogs/year). PDH typically affects dogs aged six years and older, with a predisposition observed in several breeds, including Bichon Frise, Dachshund, Yorkshire Terrier, Miniature Poodle, Irish Setter, and Basset Hound [45,46]. Since early epidemiological studies revealed a higher incidence of canine CD compared to humans, the possibility that dogs could serve as a spontaneous model for studying human CD has been considered [2]. Over the years, several studies have collected evidence about similarities between CD in dogs and humans as well as differences that may hamper direct extrapolation of findings from one species to another. One major anatomical difference concerns the organization of the pituitary gland. Although the general structure is conserved between the two species, canine ACTH-secreting cells are found not only in the anterior lobe, as in humans, but also in the intermediate zone. Consequently, canine corticotroph adenomas originate from both regions, with approximately 10% originating in the intermediate lobe [20,30].

At the clinical level, CD manifests similarly in dogs and humans, with symptoms including abdominal obesity, weight gain, fatigue, muscle atrophy, skin changes, and increased overall mortality [2,20]. However, subtle differences have been described. For instance, dogs with PDH primarily exhibit polyuria and polydipsia, symptoms not typically seen in humans. In contrast, osteoporosis and hirsutism are common in human CD patients but do not occur in dogs [2].

An important question that several studies have attempted to address is whether the genetic etiology of dogs and humans is actually the same. One challenge in answering this question arises from the increased susceptibility to tumor development due to the limited gene pool resulting from pedigree breeding [47]. Moreover, for a long time, even the human genetic basis of CD remained unclear [48]. Among the genetic similarities between corticotroph adenomas of dogs and humans, a potential molecular basis for glucocorticoid resistance has been described. Indeed, approximately 50% of both human and canine corticotroph adenomas lack nuclear expression of BRG1 and HDAC2, two factors involved in the regulation of the negative feedback of the HPA axis [19].

Activating mutations in the gene encoding ubiquitin-specific protease 8 (USP8), predominantly clustered within the 14-3-3 protein binding motif, has been proposed as a key driver of corticotroph adenoma formation in humans [49,50,51]. These mutations seem to prolong the effect of epidermal growth factor (EGF) signaling, which in turn enhances the expression of proopiomelanocortin (POMC), the ACTH precursor. Although USP8 mutations have not been identified in canine ACTH-secreting pituitary adenomas, increased nuclear expression of USP8, similar to that observed in USP8-mutated human tumors, has been reported in dogs, correlating with smaller tumor size but elevated ACTH production [20]. The potential of dogs as translational models has also been assessed through the analysis of pharmacological targets currently used in human therapy. In this context, De Bruin and collaborators examined the expression of dopamine (DA) and somatostatin (SS) receptors [52], as compounds targeting these receptors are currently used to successfully reduce ACTH and cortisol levels in subsets of human CD patients. DA and SS receptor subtypes were found to be functionally expressed in canine adenomas, with some differences compared to their human counterparts. In particular, the SS receptor subtype (sst) 2 was highly expressed in canine adenomas, whereas its expression in human adenomas is very low. Additionally, DA receptor subtype 2 (D_2_) was moderately well expressed, while sst5 expression was remarkably low, differing from the pattern observed in human adenomas. This leads to the hypothesis that dogs may not be reliable models for testing novel SS analogues and DA agonists for future use in human CD patients, as the differences in SS and DA receptor expression compared to humans could influence the pharmacological response [52]. Despite this aspect, subsequent studies demonstrated that pasireotide was capable of decreasing ACTH and adrenal cortisol levels in dogs, without causing severe adverse side effects [21,22]. These findings were later confirmed in humans [53]. Additional evidence supporting the potential translation of pharmacological findings from canine studies to humans comes from studies on retinoic acid. Although clinical studies in humans are still limited, retinoic acid appears capable of inhibiting ACTH production and reducing cortisol levels in dogs, as in humans [23,24,25]. Similarly, the epidermal growth factor receptor (EGFR) has been proposed as a potential pharmacological target for the treatment of CD in dogs and humans. Indeed, EGFR is expressed in ACTH-secreting pituitary adenomas in both species, and its inhibition with the EGFR inhibitor gefitinib has been shown to reduce POMC expression, subsequent ACTH production, and corticotroph cell proliferation [26]. Interestingly, an ongoing clinical trial is evaluating gefitinib in human CD patients with USP8 mutations, but no results have been released so far (Targeted therapy with gefitinib in patients with USP8-mutated Cushing’s disease; ClinicalTrials.gov (accessed on 13 February 2024), identifier: NCT02484755). Given the molecular conservation of EGFR and similar drug responses, canine studies could serve as valuable predictors of therapeutic efficacy in humans. Beyond the implications for human medicine, these pharmacological studies also bear clinical relevance in veterinary medicine. To date, standard treatment of canine PDH is aimed at regulating cortisol levels with drugs such as mitotane or trilostane. Hypophysectomy has been demonstrated to effectively reduce hormone levels to below normal thresholds [54,55]. Even if radiotherapy can contribute to adenoma size reduction, it seldom leads to complete remission of hypercortisolism [56,57,58]. Nevertheless, due to their limited accessibility and potential risks, these treatments are available only in a few specialized centers.

Further molecular similarity between canine and human CD is related to the expression profiles of microRNAs (miRNAs). In dogs with PDH, six miRNAs—miR-122-5p, miR-126-5p, miR-141-3p, miR-222-3p, miR-375-3p, and miR-483-3p—have been found to be differentially expressed compared to healthy controls. Among these, miR-122-5p showed significantly increased expression in PDH dogs, which markedly decreased following hypophysectomy and rose again in cases of disease recurrence, supporting its potential role as a prognostic biomarker [27]. In humans, multiple studies have reported altered expression profiles of various miRNAs in CD patients compared to healthy individuals, based on analyses of both plasma samples [59,60,61] and tumor tissue [62,63,64,65,66]. Interestingly, miR-122-5p and miR-141-3p appear to be similarly deregulated in both species. Notably, miR-122-5p is known to contribute to the regulation of cell proliferation and metastasis formation in various neoplasms, including non-small cell lung cancer [67,68], and pancreatic cancer [69,70]. Similarly, miR-141-3p plays multifunctional roles in a range of biological processes that drive tumor progression, particularly in pancreatic [71] and prostate cancer [72], suggesting a potentially conserved involvement of these miRNAs in the progression of CD across species. However, comparative functional studies will be essential to clarify whether other miRNAs, although initially appearing to be species-specific deregulated, may actually converge on shared molecular pathways, reflecting conserved biological responses relevant to human disease. Such investigations will be crucial to assess whether these miRNAs can serve as reliable, non-invasive biomarkers in both canine and human CD, ultimately advancing diagnostic strategies and facilitating the identification of novel therapeutic targets.

In addition to the well-documented molecular homologies between canine and human corticotroph pituitary adenomas, another clinical parallel lies in the frequent coexistence of hypothyroidism with CD. Thyroid dysfunction, often presenting as central hypothyroidism, has been repeatedly observed in both species, likely resulting from the suppressive effects of chronic hypercortisolism on the hypothalamic–pituitary–thyroid axis [28,29]. This shared endocrine comorbidity further supports the translational relevance of the canine model for studying the systemic impact of CD.

### 3.2. Horse (Equus caballus)

Epidemiological studies showed that Equine Cushing’s Disease (ECD), unlike the human counterpart, is a relatively common disorder in horses, with a reported prevalence of 20% to 25% and an onset around 15 years of age [36]. The expression ECD has historically been used to distinguish a clinical condition that more closely resembles human CD [73]. Nevertheless, the increasing knowledge on the underlying mechanisms in horses, which were found to be substantially different from the human counterpart, has recently led to the formulation of a new term, namely, Pituitary Pars Intermedia Dysfunction (PPID), that more accurately reflects the anatomical and functional damage that is causative of the disease [37]. In equines, differently from the human context, the direct innervation of the melanotrope cells in the intermediate pituitary by the hypothalamic dopaminergic neurons is suggestive of the important role played by dopamine, which interacts with specific receptors on these cells, inhibiting cell proliferation and transcription of POMC [36]. Even if the pathophysiology of PPID in horses has still to be completely elucidated, the formation of ACTH-secreting pituitary tumors in PPID has been attributed to progressive neurodegeneration. This leads to the loss of dopaminergic inhibition over the intermediate pituitary, with the consequent overproduction of POMC peptide hormones, including α-MSH, β-END, and β-MSH, and a moderate increase in ACTH [73,74]. Moreover, far from what happens in the other species, including humans, it has been observed that the increased secretion of ACTH is due to the more elevated rate of POMC biosynthesis rather than a preferential processing of POMC to ACTH [7,75]. Based on these premises, although ECD and PPID are commonly used as synonyms, particular attention is needed in comparative endocrinology studies, considering that PPID has a neurodegenerative component that has never been described in human CD.

Another substantial divergence between CD in humans and horses concerns the anatomical origin of pituitary tumors. Indeed, from the first description of the disease in the German veterinary literature, CD-causing hyperplasia or tumors have been observed in the intermediate lobe of the pituitary gland, while in humans, the anterior portion is more frequently involved [74]. Considering that the equine intermediate pituitary encompasses the region corresponding to the human posterior pituitary, tumors arising from this area often expand and compress surrounding tissues, including the hypothalamus. In humans, such compression is very rare in CD, as the tumors typically originate in the anterior lobe and are usually microadenomas with limited mass effect [37].

Despite anatomical and pathophysiological differences, horses affected by PPID share several similarities in clinical manifestations with humans, including hirsutism, hyperglycemia, muscle wasting, and behavioral alterations such as lethargy and narcolepsy [34]. Laminitis, a complication of chronic PPID in horses, which is usually the result of a decreased glucose uptake and insulin regulation, has common features with metabolic syndrome of CD patients. Both species are frequently subject to several opportunistic infections, likely caused by chronic immunosuppression or impaired wound healing process. In addition, abnormal fat deposition and excessive adipose tissue in specific areas are suggestive of a condition that partially resembles weight alterations and obesity characterizing human CD [37]. Whereas in humans, the increase in abdominal adipose tissue is more frequent, in equine patients, it concentrates along the neck and over the tail head, in the sheath of the males, and at the level of the supraorbital area. Interestingly, despite no sex predilection being documented in horses, distinctive signs of PPID include the presence of persistent lactation and infertility in mares, reflecting the hypothalamic–pituitary–gonadal axis dysregulation described in women affected by CD [32,33,34,76].

Another aspect that horses share with humans is the expression of dopamine receptors in CD-causing pituitary tumors, and, in this frame, a similar response to dopamine agonists has been observed in both species [35]. The only registered drug for the therapy of PPID-affected horses is pergolide [31,77], a D_2_ receptor agonist that significantly improves clinical signs by reducing ACTH concentrations within the accepted intervals in 28–74% of cases [31]. This therapeutic profile closely reflects those exerted by cabergoline, or other dopaminergic drugs, which have been proposed as alternative treatments for persistent or recurrent forms of CD in humans [78]. Indeed, considering that a significant percentage of human corticotropic adenomas responsible for CD and several extra-pituitary tumors express D_2_ dopamine receptors, which play a key role in the control of ACTH secretion and tumor growth, cabergoline and other dopaminergic drugs can be used as first therapeutic options or adjuvants in the case of surgery contraindications or refusal. These drugs can help reduce ACTH secretion and, in some cases, even lead to tumor shrinkage [78]. Altogether, these data suggest a translational potential of PPID in the preclinical assessment of novel dopamine receptor agonists, which therefore include long-acting compounds.

### 3.3. Cat (Felis catus)

Feline CD, more commonly referred to as PDH in veterinary medicine, accounts for approximately 80% of all described cases of feline hyperadrenocorticism [79]. Despite being rarer than in dogs and relatively uncommon even in humans, feline CD shares several pathophysiological and clinical features with its human counterpart, offering potential insights into translational aspects of the disease [38]. As in dogs, feline PDH is typically caused by corticotroph adenomas arising from both anterior and intermediate lobes of the pituitary gland [39]. Muscle wasting and generalized weakness are among the most consistent clinical features shared by cats and humans, both arising from the sustained catabolic effects of glucocorticoids on protein metabolism. These manifestations are often accompanied by lethargy and reduced physical activity. Another major point of convergence is GC-induced insulin resistance, which frequently progresses to diabetes mellitus. In cats, this metabolic complication is particularly prevalent and often dominates the clinical picture. Diabetes, in turn, contributes to the development of polyuria and polydipsia. Visceral fat redistribution is also observed in both species, which typically is manifested through abdominal distension [38].

One particularly intriguing aspect of feline CD, with possible translational relevance, is the apparent glucocorticoid resistance observed in clinically normal cats. This phenomenon is supported by evidence indicating that cats exhibit species-specific characteristics, possessing a low density of GC receptors with a reduced binding affinity [80]. However, considering the lack of genetic knowledge on corticotroph tumors, further studies are needed to assess whether the feline model can be reliably used to explore the mechanisms of GCs resistance and its implications in human CD.

### 3.4. Small Mammals

Our bibliographic research on spontaneous forms of CD has also been extended to small mammalian species that, although less commonly used, are occasionally employed in biomedical studies. In Guinea pigs (*Cavia porcellus*) as well as in different hamster species such as *Mesocricetus auratus* and *Cricetus cricetus*, a few cases of hyperadrenocorticism associated with pituitary tumors have been described. Among the reported symptoms, the most relevant parallels with humans are dermatological manifestations and muscle weakness, while polydipsia and polyuria may also occur [40,41,42]. However, these studies, primarily of veterinary interest in companion animals, have not suggested the use of such spontaneous forms of CD as models for research with translational implications. Indeed, information on their incidence remains inconsistent. Moreover, the use of these species in modern laboratories is limited, as they are larger and more expensive to house than mice, and fewer standardized reagents, protocols, and genetic tools are available.

### 3.5. Rat (Rattus norvegicus)

Although they also belong to the already treated category of the “small mammals”, rats deserve a separate discussion, since they are part of a species more widely employed in biomedical research. Spontaneous forms of CD have been identified in rats, mainly due to ACTH-secreting pituitary adenomas. Reported symptoms, which include severe obesity, inflammation in multiple organs, fertility impairment, dermatological alterations, and muscle weakness, closely resemble the human CD phenotypes [43,44]. Nevertheless, no reliable data on incidence are currently available, which prevents the potential use of these spontaneous cases in rats as a consistent model for biomedical research.

## 4. CD Animal Models for Preclinical Research

Among the animal species routinely used in preclinical research, models for the study of CD have been generated in mice, rats, and zebrafish. As with other tumor types, both genetically engineered models, which exploit DNA modification technologies to recreate the molecular conditions underlying the disease, and xenograft models, based on the transplantation of tumor cells, have been established in these species to study the pathogenetic basis of CD, the systemic effects of hypercortisolism, and to promote the development of novel therapeutic strategies. CD models for preclinical research of CD are summarized in Table 3.

### 4.1. Mouse (Mus musculus)

One of the experimental strategies adopted in mice to investigate the pathophysiology of CD was based on the generation of various transgenic lines overexpressing the CRH gene. Although the formation of ACTH-secreting pituitary tumors has not been observed in these models, several CD-related phenotypes have been successfully recapitulated. In the first transgenic model, CRH was ubiquitously expressed under the control of the metallothionein-1 (MT1) promoter [81]. These mice displayed hallmark features of CD, including anxiety-like behavior, central obesity, osteoporosis, and muscle weakness, which were associated with elevated circulating levels of ACTH and corticosterone. Another approach employed the Thy1 promoter to drive constitutive CRH expression in neurons throughout embryonic development and adulthood. In this context, animals exhibited alterations in stress response, such as reduced startle reactivity and diminished freezing behavior in fear-conditioning paradigms, along with a mild hypercortisolemic phenotype [82,83,84]. Conditional models were also developed to achieve systemic or region-specific overexpression of CRH. In particular, mice engineered to express CRH either throughout the entire body or specifically within the anterior and intermediate lobes of the pituitary both displayed increased baseline plasma corticosterone concentrations and signs of CD. Notably, generalized CRH overexpression led to increased anxiety-related behaviors, whereas pituitary-specific overexpression did not appear to impact emotional reactivity [85]. A separate model, generated through ENU mutagenesis, carried a point mutation located −120 bp upstream of the *Crh* promoter, resulting in the constitutive activation of the gene and chronic stimulation of the HPA axis [86]. These mice exhibited a CD-like phenotype characterized by visceral obesity, hyperglycemia and muscle wasting, reduced bone mineral density, and elevated plasma corticosterone levels. Furthermore, transgenic mouse lines expressing the polyoma large T (PyLT) antigen have been reported to develop ACTH-producing pituitary tumors and late-onset CD-like symptoms, including weight gain, adrenal medullary hyperplasia, and increased peripheral ACTH levels [87,88]. Based on these phenotypic characteristics, these models were proposed as representative of CD. However, another research group used the same PyLT construct without observing the presence of pituitary adenomas and CD-like symptoms [104]. This discrepancy underscores a potential limitation of the PyLT transgenic approach and highlights the variability in phenotypic outcomes, even when identical genetic constructs are used.

An effective strategy to reproduce key features of CD in mice has been the xenotransplantation of ACTH-producing cells. The first pioneering study in this field dates back to 1992 and involved the subcutaneous implantation of AtT-20 cells, a murine pituitary corticotroph tumor cell line, into immunodeficient mice [89]. These cells rapidly formed tumors, and the recipient mice developed a phenotype consistent with CD. Six weeks post-implantation, the mice showed a 45% increase in body weight compared to controls, primarily due to excessive fat accumulation. A characteristic “buffalo hump” appearance, caused by the abnormal redistribution of adipose tissue, was observed. Additionally, histological analyses revealed adrenal gland enlargement attributable to zona fasciculata hypertrophy [89]. This transplantable model has been widely used for preclinical drug testing, particularly to assess the therapeutic efficacy of agents targeting CD-related pathways. Compounds tested in this model include pasireotide [90] and new possible treatments for CD, such as parthenolide [91], gefitinib [26], silibinin [92], suberoylanilide hydroxamic acid [93], thiostrepton [94], metformin [95], bexarotene [96], and triptolide [97].

More recently, a modified cell line, AtT-20/D16v.2, has been utilized for subcutaneous implantation in mice, successfully inducing phenotypes resembling CD as weight gain and fat deposition [98]. This variant has been used to study the effects of PD168368, a Neuromedin B receptor antagonist, on tumor growth and hormonal output [99]. The potential advantages of using the D16v.2 subclone, such as more consistent tumor take or enhanced ACTH production, are currently under investigation.

Despite the extensive use of xenograft models derived from AtT-20 cells, the establishment of patient-derived xenografts (PDX) from ACTH-secreting pituitary adenomas remains elusive. Unlike other tumor types, where PDXs have become indispensable tools in translational oncology, their application in CD research is limited by several factors. These include the benign and slow-growing nature of corticotroph tumors, the small size of surgical specimens, and the critical dependence of tumor cells on the highly specialized pituitary microenvironment, which is difficult to recapitulate in xenograft settings. To date, no robust PDX model of ACTH-producing tumors has been successfully developed in mice.

### 4.2. Rat (Rattus norvegicus)

Unlike the mouse, no genetic models of CD are currently available in rats. With regard to xenograft-based approaches, although some studies have reported the injection of ACTH-secreting tumor cells into rats, the results have not provided promising perspectives for the study of CD. In an early work, human ACTH-secreting pituitary tumor cells were injected into the pituitary fossa of total-body irradiated, hypophysectomized rats with the aim of identifying potential environmental factors influencing the survival of implanted cells [105]. In another study, AtT-20 cells, which produce β-endorphin, and AtT-20/hENK cells, which are AtT-20 cells transfected with a proenkephalin gene, were implanted into the rat spinal subarachnoid space in an attempt to explore a therapeutic approach for the treatment of chronic pain [106].

More recently, a CD platform was developed by the injection of CRH (0.5 mg/kg) in Sprague–Dawley rats, which induced the secretion of ACTH and the subsequent increase in corticosterone levels. This platform was used to demonstrate the higher inhibitory effect of the multireceptor ligand SOM230, compared with the sst2-preferring agonist octreotide, on corticotropin-releasing hormone (CRH)-stimulated secretion of ACTH and corticosterone [100].

Similarly, the typical CD hypercortisolism has been developed on Sprague–Dawley rats based on ACTH infusion (40 ng/day) via osmotic minipumps for four weeks. This experimental model has been employed to investigate the molecular basis of muscle weakness in CD, identifying a role for the forkhead box3a in muscle atrophy through the regulation of muscle RING finger protein-1 and atrogin-1 expression [107]. Furthermore, this CD model was found to be suitable to identify the histone deacetylase inhibition as a possible pharmacological strategy to ameliorate hypertension, hyperglycemia, and hepatic steatosis induced by ACTH-dependent GC excess [101,102].

### 4.3. Zebrafish (Danio rerio)

Despite the evolutionary distance from humans, zebrafish represent a powerful experimental model with unique features to support the development of translational models for CD. Zebrafish has gained increasing attention in biomedical research due to several intrinsic advantages such as the high prolificity and the low costs of maintenance, together with the external fertilization and transparency of its embryos that allow the in vivo observation of developmental processes. Furthermore, the high permeability to small molecules of embryos and larvae facilitates compound screening, making zebrafish a particularly suitable organism for high-throughput drug discovery [108,109,110]. Importantly, approximately 70% of human protein-coding genes have at least one orthologue in zebrafish, and many biological functions are conserved [111].

In the context of CD, a key translational feature of zebrafish is the conservation of many neuroendocrine pathways between teleosts and mammals. Zebrafish possess a homologous counterpart to the mammalian HPA axis, which is called the hypothalamic–pituitary–interrenal (HPI) axis. The interrenal tissue in zebrafish acts similarly to the adrenal glands in mammals. Notably, cortisol is the primary glucocorticoid in both humans and zebrafish, unlike rodents that use corticosterone. In zebrafish, cortisol binds to GC receptors (GRs), encoded by a single gene with two splice variants, similarly to humans [112]. Interestingly, the HPI axis in zebrafish is already active during embryonic development, offering a unique opportunity to study the effects of genetic or pharmacological manipulations at early stages of development. Notably, it has been shown that *pomc*, the gene encoding the precursor of ACTH, is first expressed asymmetrically as two bilateral clusters of cells located anterior to the neural ridge midline at approximately 18–20 h post-fertilization (hpf). By 24 hpf, *pomc*-expressing cells converge to form a single-cell mass within the pituitary anlage, and between 48 and 64 hpf, they spatially organize into distinct anterior and posterior domains within the developing pituitary [113]. In parallel, the mRNA levels of key steroidogenic enzymes, such as steroidogenic acute regulatory protein, 11β-hydroxylase, and 11β-hydroxysteroid dehydrogenase type 2, increase significantly after hatching, preceding the detectable rise in endogenous cortisol levels [114].

One notable and relevant example for CD is the transgenic model developed by Liu and collaborators (2011) overexpressing the pituitary tumor-transforming gene (PTTG/securin), a gene often dysregulated in pituitary tumors, in *pomc*-positive pituitary lineages [103]. Interestingly, adult transgenic fish developed neoplastic corticotrophs and metabolic disturbances mimicking the typical hypercortisolism state of CD. However, the most interesting aspect of this transgenic line was the presence of corticotroph expansion with cyclin E upregulation and cell-cycle dysregulation in embryonic and larval stages, together with a partial GC resistance. Thus, the early onset of corticotroph adenoma conditions offers the unique opportunity to exploit the proven advantages of zebrafish embryos and larvae as a platform for in vivo drug testing. In this frame, the Tg:Pomc-Pttg model has been used to evaluate the efficacy of R-roscovitine, a cyclin-dependent kinase (CDK) inhibitor, which showed a strong activity in suppressing ACTH expression, inducing senescence in corticotroph tumor cells, and downregulating cyclin E [103].

The versatility of zebrafish also allows exploration of one of the CD-related comorbidities, reproductive dysfunction. In 2015, Sousa and collaborators demonstrated differential responses to cortisol and ACTH in zebrafish ovarian follicles, suggesting a direct impact of stress signaling on female gametogenesis [115]. These findings underscore the potential of zebrafish to investigate the interplay between the stress axis and reproductive biology, offering new insights into the systemic consequences of CD.

## 5. Conclusions

To date, many preclinical studies on CD have been conducted in vitro using 2D cell models, although only a limited number of ACTH-secreting lines are available. The AtT-20 mouse corticotroph cell line, established in the 1950s, together with its subclone AtT-20/D16v.2, has been extensively used to investigate *Pomc* regulation and to screen potential therapeutics [19,116]. DMS79 is the only human immortalized cell line currently used for CD research [117]. However, it is not of pituitary origin, but derived from human small cell lung cancer. In this cell line, secretion of unprocessed POMC is favored over mature ACTH [118,119], and important differences in POMC regulation compared to AtT-20 cells have been reported. More recently, the emergence of human pituitary organoids has extended the viability of primary cultures up to several weeks and offers the possibility to recapitulate the complex interactions involved in pituitary development [120,121,122]. Altogether, these advances in cellular modeling represent important steps toward capturing the molecular complexity of CD in vitro. Nevertheless, given their intrinsic limitations, it remains essential to further identify or develop animal models capable of reproducing the distinctive pathophysiological features of CD, thereby complementing cellular systems and providing a more integrated platform for translational research.

Spontaneously occurring CD in dogs, horses, and cats offers valuable opportunities for translational research, owing to notable physiological and molecular similarities with the human condition. Among these species, the dog has emerged as the most thoroughly studied model, with compelling evidence of conserved molecular pathways, pituitary tumorigenesis mechanisms, and systemic comorbidities. Equine and feline cases, though less extensively characterized, also exhibit features that reflect the human disease, particularly in terms of chronic hypercortisolism and GC resistance. However, the use of these companion animal models presents practical and ethical challenges, including variability in clinical presentation, limited availability of standardized genetic tools, and regulatory constraints that may hinder experimental manipulation. Compared to classical laboratory models, such as mouse or zebrafish, which offer unparalleled genetic tractability and experimental reproducibility, companion animals are less amenable to high-throughput or mechanistic studies. Nonetheless, their unique value lies in modeling the spontaneous, long-term progression of CD in a physiologically relevant context, making them complementary, rather than alternative, to laboratory traditional systems.

Preclinical models of CD developed in mice, rats, and zebrafish have significantly contributed to the characterization of various pathological features and to the testing of novel therapeutic strategies. Most of the research so far has been conducted in mice, using both stable transgenic models and xenograft-based models with ACTH-secreting tumor cells. However, stable mouse lines with a proven predisposition to develop ACTH-secreting tumors are currently unavailable. This represents a limitation in the standardization of preclinical studies in CD. Compared to mice, rats and zebrafish have been less frequently used to model CD. Rat studies, in which key features of CD were experimentally induced through the administration of CRH [100] or ACTH [101,102,107], have provided valuable preliminary insights into the molecular basis of the disease and supported the development of novel pharmacological strategies. Although this experimental approach has proven useful for reproducing characteristic manifestations of CD, such as hypertension, hyperglycemia, and muscle weakness, it does not fully recapitulate the complexity of the clinical phenotype. Regarding zebrafish, the available transgenic line offers notable advantages for CD studies. In particular, intrinsic features of this experimental model allow the investigation of disease-related processes from the larval stages, when the optical transparency of embryos enables real-time in vivo imaging. Moreover, adult Tg:Pomc-Pttg zebrafish exhibit hypercortisolism, closely resembling the human disease phenotype [103]. Given the well-established value of zebrafish in preclinical research, this model holds promise for further applications aimed at elucidating the pathogenic mechanisms and progression of CD, as well as for the development of new pharmacological approaches.

One promising avenue involves the use of zebrafish for tumor cell xenografts. Numerous studies have highlighted the advantages of zebrafish xenografts over murine models, particularly when performed in embryos and larvae. These benefits include a significantly shorter time to tumor engraftment (days rather than weeks or months), the absence of a requirement for immunosuppression, a higher feasibility of modeling metastasis, and the need for a much smaller number of tumor cells (often fewer than one million). This last feature is particularly relevant to CD research, where PDX models are lacking, mainly due to the extremely limited quantity of tissue available following surgical resection of pituitary or ectopic ACTH-secreting tumors. Zebrafish PDX models may offer a unique opportunity to overcome this limitation and facilitate personalized medicine applications by enabling the testing of individualized therapeutic responses in vivo. Furthermore, zebrafish have proven effective as biosensors for tumor-derived signals. In this context, several studies have exploited the model to examine tumor-induced angiogenesis and innate immune responses in embryos. The ability to generate transgenic lines expressing reporter genes under the control of specific stimuli could further enhance the utility of zebrafish for studying in vivo dysregulation of the hypothalamic–pituitary–adrenal (HPA) axis, a hallmark of CD.

## Figures and Tables

**Table 1 ijms-26-08626-t001:** Clinical manifestations of human CD.

Category	Clinical Manifestations
**Physical signs**	Moon face, buffalo hump (excessive fat accumulation on the upper back), central obesity, limb muscle wasting, and wide purple striae.
**Cardiovascular/** **Metabolic disorders**	Hypertension, glucose intolerance, dyslipidemia.
**Endocrine/** **Reproductive disturbances**	Amenorrhea, infertility, impaired growth in pediatric patients.
**Musculoskeletal complications**	Osteoporosis, proximal muscle weakness, reduced mobility, increased fracture risk.
**Neuropsychiatric disorders**	Depression, anxiety, reduced health-related quality of life.
**Immune/** **Hematologic complications**	Immunosuppression, venous thromboembolism.
**Dermatological manifestations**	Acne, skin thinning, subcutaneous bleeding, pigmentation changes, hirsutism.

**Table 2 ijms-26-08626-t002:** Spontaneous forms of CD in animals.

Species	Name of Spontaneous CD Form	Similarities with Human CD	Differences with Human CD	Possible Future Research with Translational Implications for Human CD
**Dog (*Canis lupus familiaris*)**	Pituitary-dependent hyperadrenocorticism	Corticotroph adenomas lack nuclear expression of BRG1 and HDAC2 [19] and overexpressed elevated nuclear USP8 protein [20]. Pasireotide reduced ACTH and cortisol levels, despite differences in SS and DA receptor expression [21,22]. RA reduces ACTH and cortisol levels [23,24,25]. EGFR inhibitors, reduce POMC expression, ACTH production, and corticotroph cell proliferation [26]. Changes in miRNA expression compared to healthy subjects (i.e., miR-122-5p and miR-141-3p) [27]. Frequent coexistence of hypothyroidism with CD [28,29].	High incidence (1 to 2 cases/1000 dogs/year) [2]. Corticotroph adenomas originate from both anterior lobe and intermediate zone of pituitary gland [20,30]. Subtle differences in symptoms [2]. Subtle differences in the expression of SS and DA receptors [2].	Possible prediction of the therapeutic efficacy of new SS/DA compounds, despite moderate differences in the SS/DA expression profiles between humans and dogs. Possible prediction of the therapeutic efficacy of new EGFR inhibitors. Evaluation of miRNA reliability as non-invasive biomarkers for CD. Characterization of the systemic impact of CD, with a particular focus on thyroid.
**Horse (*Equus caballus*)**	Equine Cushing’s Disease or Pituitary Pars Intermedia Dysfunction	Dopaminergic system modulates the HPA axis, both under normal and pathological circumstances [31]. Several similarities in clinical manifestations [32,33,34]. Dopamine agonists reduce ACTH secretion and improve prognosis [31,35].	Common endocrine disorder with a prevalence of 20–25% [36]. ACTH-secreting pituitary tumors form as a result of a progressive neurodegeneration that causes the loss of dopaminergic inhibition over the intermediate pituitary [37]. Corticotroph adenomas originate from the intermediate zone of pituitary gland [37].	Evaluation of the therapeutic efficacy of dopamine agonists.
**Cat (*Felis catus*)**	Pituitary-dependent hyperadrenocorticism	Rare incidence [38]. Muscle wasting and generalized weakness [38]. Lethargy and reduced physical activity [38]. Visceral fat redistribution, typically through abdominal distension [38].	Corticotroph adenomas originate from both anterior lobe and intermediate zone of pituitary gland [39].	Characterization of the resistance to corticosteroids.
**Small mammals** **Guinea pigs (*Cavia porcellus*)** **Hamster (*Mesocricetus auratus* and *Cricetus cricetus*)**	Pituitary-dependent hyperadrenocorticism	Dermatological manifestations [40,41,42]. Muscle weakness [40,41,42].	Polydipsia and polyuria [40,41,42].	Inconsistent data to support future studies.
**Rat (*Rattus norvegicus*)**	Pituitary-dependent hyperadrenocorticism	Severe obesity [43,44]. Inflammation in multiple organs [43,44]. Fertility impairment [43,44]. Dermatological alterations [43,44]. Muscle weakness [43,44].	Unavailable data.	Inconsistent data to support future studies.

Abbreviations: ACTH, adrenocorticotropic hormone; CD, Cushing’s Disease; DA, dopamine; POMC, proopiomelanocortin; RA, retinoic acid; SS, somatostatin.

**Table 3 ijms-26-08626-t003:** CD animal models for preclinical research.

Species	Experimental Strategy	CD Features
**Mouse** **(*Mus musculus*)**	Transgenic CRH expression under the control of the metallothionein-1 (MT1) promoter [81]. Transgenic CRH expression under the control of the Thy1 promoter to drive constitutive expression in neurons throughout embryonic development and adulthood [82,83,84]. Conditional overexpression of CRH throughout the entire body or within the anterior and intermediate lobes of the pituitary [85]. Point mutation located −120 bp upstream of the Crh promoter through ENU mutagenesis [86]. Transgenic mouse lines expressing the polyoma large T (PyLT) antigen [87,88]. Subcutaneous implantation of AtT-20 cells into immunodeficient mice [26,89,90,91,92,93,94,95,96,97]. Subcutaneous implantation of AtT-20/D16v.2 into immunodeficient mice [98,99].	Anxiety-like behavior, central obesity, osteoporosis, and muscle weakness, which were associated with elevated circulating levels of ACTH and corticosterone. Alterations in stress response along with a mild hypercortisolemic phenotype. Increased baseline plasma corticosterone concentrations, anxiety-related behaviors only occurred when CRH was ubiquitously overexpressed. Chronic stimulation of the HPA axis, visceral obesity, hyperglycemia and muscle wasting, reduced bone mineral density, and elevated plasma corticosterone levels. Development of ACTH-producing pituitary tumors, weight gain, adrenal medullary hyperplasia, and increased peripheral ACTH levels. Increase in body weight, excessive fat accumulation, “buffalo hump” appearance, adrenal gland enlargement. Weight gain and fat deposition.
**Rat** **(*Rattus*** ***norvegicus*)**	Injection of CRH (0.5 mg/kg) [100]. Infusion of ACTH (40 ng/day) [101,102].	Stimulation of ACTH secretion and subsequent increase in corticosterone levels. Increase in corticosterone levels, hepatic steatosis, hypertension, and hyperglycemia.
**Zebrafish** **(*Danio rerio*)**	Transgenic overexpression of the pituitary tumor-transforming gene (PTTG/securin) in POMC-positive pituitary lineages [103].	Development of neoplastic corticotrophs and metabolic disturbances mimicking the typical hypercortisolism state in adult fish. Presence of corticotroph expansion with cyclin E upregulation and cell-cycle dysregulation in embryonic and larval stages, together with a partial GC resistance.

Abbreviations: ACTH; adrenocorticotropic hormone; CRH, corticotropin-releasing hormone; ENU, N-ethyl-N-nitrosourea; GC, glucocorticoid; HPA, hypothalamic–pituitary–adrenal axis.

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
