# Peer review of "Cushing’s Disease in the Animal Kingdom: Translational Insights for Human Medicine"

_ijms, 2025, doi:10.3390/ijms26178626_

Round 1

Reviewer 1 Report

Comments and Suggestions for Authors

Comments for Authors:

The review article titled “Cushing’s Disease in the Animal Kingdom: Translational Insights for Human Medicine” is a comprehensive and well-structured thematic review that explores the potential translational value of using diverse experimental designs, from domestic and/or genetically modified animals to contribute to the understanding of an extremely important endocrine pathology: Cushing’s disease, a neuroendocrine disorder associated with ACTH-secreting pituitary adenomas.

Taking into account a highly significant feature, such as the conservation of the hypothalamic-pituitary-adrenal axis at the evolutionary level, particularly in the species evaluated, the authors present varied and alternative experimental systems for addressing Cushing’s disease.

Throughout the manuscript, different possible experimental models and approaches for developing this endocrine pathology are described, particularly focusing on domestic animals, elaborating on their strengths and weaknesses, as well as their potential translational value.

The authors present, in depth, the clinical and the review examines the molecular mechanisms of dogs, horses, and cats with this human pathology, presenting the ethical and practical challenges that the use of these animal models in laboratory practice would entail.

In addition to elaborating on companion animal species, the review emphasizes the importance of using mouse and zebrafish models as preclinical models based on their genetic characteristics, particularly with regard to the evolutionary conservation of neuroendocrine pathways. This is not a minor aspect, since resorting to the use of species evolutionarily distant from humans but which nevertheless retain genetic and clinical similarity to humans represents enormous added value for the study of Cushing's disease.

The main contribution of this review lies in showing how a deep understanding of this endocrine pathology in animals, both in spontaneous and experimental conditions, can contribute to the field of human health, not only in understanding but also in diagnosing and developing therapies for this complex pathology in humans, considering its high impact on the field of psychiatry.

In addition to the clear and fluid presentation, the manuscript contains a wealth of information. A bibliography that includes foundational articles on the use of companion animals to understand this endocrine disease, as well as updated reports.

In short, this is a solid work and constitutes an additional contribution to the understanding of this common pathology in the field of human endocrinology.

Author Response

COMMENT: The review article titled “Cushing’s Disease in the Animal Kingdom: Translational Insights for Human Medicine” is a comprehensive and well-structured thematic review that explores the potential translational value of using diverse experimental designs, from domestic and/or genetically modified animals to contribute to the understanding of an extremely important endocrine pathology: Cushing’s disease, a neuroendocrine disorder associated with ACTH-secreting pituitary adenomas.

Taking into account a highly significant feature, such as the conservation of the hypothalamic-pituitary-adrenal axis at the evolutionary level, particularly in the species evaluated, the authors present varied and alternative experimental systems for addressing Cushing’s disease.

Throughout the manuscript, different possible experimental models and approaches for developing this endocrine pathology are described, particularly focusing on domestic animals, elaborating on their strengths and weaknesses, as well as their potential translational value.

The authors present, in depth, the clinical and the review examines the molecular mechanisms of dogs, horses, and cats with this human pathology, presenting the ethical and practical challenges that the use of these animal models in laboratory practice would entail.

In addition to elaborating on companion animal species, the review emphasizes the importance of using mouse and zebrafish models as preclinical models based on their genetic characteristics, particularly with regard to the evolutionary conservation of neuroendocrine pathways. This is not a minor aspect, since resorting to the use of species evolutionarily distant from humans but which nevertheless retain genetic and clinical similarity to humans represents enormous added value for the study of Cushing's disease.

The main contribution of this review lies in showing how a deep understanding of this endocrine pathology in animals, both in spontaneous and experimental conditions, can contribute to the field of human health, not only in understanding but also in diagnosing and developing therapies for this complex pathology in humans, considering its high impact on the field of psychiatry.

In addition to the clear and fluid presentation, the manuscript contains a wealth of information. A bibliography that includes foundational articles on the use of companion animals to understand this endocrine disease, as well as updated reports.

In short, this is a solid work and constitutes an additional contribution to the understanding of this common pathology in the field of human endocrinology.

RESPONSE: We would like to sincerely thank the Reviewer for the thorough and positive evaluation of our manuscript and for highlighting its strengths in terms of structure, clarity, and translational relevance. We greatly appreciate the recognition of our effort to integrate both spontaneous and experimental animal models in order to provide a comprehensive overview of their value for advancing the understanding of Cushing’s disease in humans. No major modifications were required; however, we carefully reviewed the manuscript once again to ensure that the presentation is as clear and consistent as possible.

Reviewer 2 Report

Comments and Suggestions for Authors

The article provides an overview of the current state of the problem of the prevalence of Cushing's disease among animals, the similarities and differences of symptoms with humans, and some preclinical models of this pathology. The article is distinguished by a good style of presentation, however, there are several comments.

In general: the analyzed databases of medical literature, the retrospective of the study, and the search keywords are not specified. I would like to see more preclinical variations of Cushing's disease modeling, for example, in rats or guinea pigs. You can also point out the presence of some cellular models. This information should also be included in the review. This will increase its quality and the possibility of citation by specialists working with small laboratory animals. 

In detail, there are not many comments: Line 76. Hypertension is not a metabolic risk. Take out cardiovascular diseases separately.

Replace the text from line 63 to line 88 with the table.

Line 102 - provide a link to the definition and its interpretation.

Lines 361-370 - give either the name of the drug or its group. Avoid the mixed variant.

Author Response

COMMENT 1: The article provides an overview of the current state of the problem of the prevalence of Cushing's disease among animals, the similarities and differences of symptoms with humans, and some preclinical models of this pathology. The article is distinguished by a good style of presentation, however, there are several comments. In general: the analyzed databases of medical literature, the retrospective of the study, and the search keywords are not specified.

RESPONSE 1: We appreciate the Reviewer’s suggestion. We have now included a paragraph in the Introduction describing the databases consulted (PubMed/MEDLINE) and the search strategy, including the main keywords used. This addition clarifies the retrospective process and increases the transparency of our review.

COMMENT 2: I would like to see more preclinical variations of Cushing's disease modeling, for example, in rats or guinea pigs. You can also point out the presence of some cellular models. This information should also be included in the review. This will increase its quality and the possibility of citation by specialists working with small laboratory animals.

RESPONSE 2: We thank the Reviewer for these important observations. We have now included additional studies about guinea pigs, hamsters and rats. This enriches the spectrum of experimental models discussed, making the review more useful for specialists of small laboratory animals. Regarding cellular models, a comprehensive review that extensively discusses the currently available 2D and 3D models for Cushing’s disease has been recently published (Hashmi H, et al. Pituitary. 2025 Apr 5;28(2):47. doi: 10.1007/s11102-025-01516-1. PMID: 40186634). In order to avoid unnecessary overlap, but still follow the reviewer’s valuable suggestion, we have added a brief mention of the advantages and limitations of these cellular models in the first part of the conclusion. This addition aims to provide a broader overview of the models currently available, highlighting their complementarity with animal models.

COMMENT 3: In detail, there are not many comments: Line 76. Hypertension is not a metabolic risk. Take out cardiovascular diseases separately.

RESPONSE 3:  We have revised the text accordingly and included a table (please see reply to comment 4).

COMMENT 4: Replace the text from line 63 to line 88 with the table.

RESPONSE 4: We thank the Reviewer for this valuable suggestion. We have transformed this section into a summary table to improve readability and provide a clearer overview of the information.

COMMENT 5: Line 102 - provide a link to the definition and its interpretation.

RESPONSE 5: We have added the appropriate reference, as suggested.

COMMENT 6: Lines 361-370 - give either the name of the drug or its group. Avoid the mixed variant.

RESPONSE 6: We have revised this section by indicating the drug name.